# GNNAS-Dock: Budget Aware Algorithm Selection with Graph Neural Networks for Molecular Docking

## Abstract

Molecular docking is a major element in drug discovery and design. It enables the prediction of ligand-protein interactions by simulating the binding of small molecules to proteins. Despite the availability of numerous docking algorithms, there is no single algorithm consistently outperforms the others across a diverse set of docking scenarios. This paper introduces GNNAS-Dock, a novel Graph Neural Network (GNN)-based automated algorithm selection system for molecular docking in blind docking situations. GNNs are accommodated to process the complex structural data of both ligands and proteins. They benefit from the inherent graph-like properties to predict the performance of various docking algorithms under different conditions. The present study pursues two main objectives: **1)** predict the performance of each candidate docking algorithm, in terms of Root Mean Square Deviation (RMSD), thereby identifying the most accurate method for specific scenarios; and **2)** choose the best computationally efficient docking algorithm for each docking case, aiming to reduce the time required for docking while maintaining high accuracy. We validate our approach on PDBBind 2020 refined set, which contains about 5,300 pairs of protein-ligand complexes. Our strategy is performed across a portfolio of 6 different state-of-the-art docking algorithms. To be specific, the candidate algorithms are DiffDock, DSDP, TankBind, GNINA, SMINA, Qvina-W. We additionally combine p2rank with GNINA, SMINA and Qvina-W for docking site prediction. Therefore, there are totally 9 different algorithms for selection. Our algorithm selection model achieves a mean RMSD of approximately 1.74 Å, significantly improving upon the top performing docking algorithm (DiffDock), which has a mean RMSD of 2.95 Å. Moreover, when making selection in consideration of computational efficiency, our model demonstrates a success rate of 79.73% in achieving an RMSD below the 2 Å threshold, with a mean RMSD value of 2.75 Å and an average processing time of about 29.05 seconds per instance. In contrast, the remaining docking algorithms like TankBind, though faster with a processing time of merely 0.03 seconds per instance, only achieve an RMSD below the 2 Å threshold in less than 60% of cases. These findings demonstrate the capability of GNN-based algorithm selection to significantly enhance docking performance while effectively reducing the computational time required, balancing efficiency with precision in molecular docking.

## 1 Introduction

Molecular docking (Stanzione et al., 2021) is critical for identifying potential drug candidates concerning computer-aided drug discovery (CADD). It has also been widely used for understanding enzyme-substrate interactions and designing enzyme inhibitors (Rudnitskaya et al., 2010). Developed computational strategies simulate the interaction between a small molecule, i.e. ligand, and a protein, i.e. receptor, at the atomic level to predict the molecule's orientation and binding affinity toward the protein. There have been a wide range of molecular docking algorithms and tools. Initial and traditional ones employ a search algorithm to determine the potential binding poses of ligands, and a scoring function to evaluate these predictions (Halperin et al., 2002). The search algorithms come into play when exploring different poses with the aim finding an acceptable, (near-)optimal

one. Those predetermined search methods and scoring functions have been gradually replaced by machine learning based techniques, offering higher generalization, flexibility and practicality. In particular, deep learning has been taking over due it its success on almost any application domains. Convolutional Neural Networks (CNN) have been frequently utilized due to the visual characteristics of the docking components, such as replacing those traditional scoring functions (McNutt et al., 2021). As an alternative example to the search-based approaches, SE(3)-equivariant geometric neural network has been introduced to predict potential binding sites and ligand orientations directly (Stärk et al., 2022), without any explicit search during prediction.

However, the diversity of available algorithms presents a challenge: no single method consistently outperforms others across all docking scenarios. This is aligned with the no free lunch theorem (NFLT) (Wolpert & Macready, 1997), which states that no single algorithm can universally excel across all problem instances. This aspect is also valid for docking algorithms, meaning that there is no ultimately best docking algorithm outperforming other under the same or experimental conditions. Thus, specifying the best docking algorithm for a docking scenario can offer significant performance gains over utilizing only one algorithm for addressing all the docking tasks at hand. Algorithm Selection (AS) (Kerschke et al., 2019) is an automated strategy to automatically accommodate one or more algorithms to solve a particular problem instance instead of relying on a single one or choosing an algorithm based on personal experience.

Targeting molecular docking, the present article proposes a Graph Neural Networks (GNNs)-based AS system, GNNAS-Dock, that utilizes structural features of ligands and proteins to predict which docking algorithm will perform optimally under varying docking conditions. GNNs are particularly well-suited for this task as they operate on graph data (Zhou et al., 2020), which naturally corresponds to the structural representation of molecules and proteins. Graph data consists of multiple nodes and edges connecting them, similar to the molecular structures composed of various atoms and the bonds between them. This structual similarity might be helpful for GNN to capture the 3D features of ligands and proteins, and predict the performance of docking algorithms on different docking tasks accurately based on these learned features. In this research, we aim to develop two distinct GNN-based algorithm selection models, each serving a specific purpose. The first model is designed to select the most accurate docking method for various scenarios. It will predict the performance of different docking methods in terms of ligand root mean square error (RMSD), and choose the one with the best predicted performance. The second model focuses on optimizing the efficiency of the docking process. It aims to reduce the time required to achieve accurate docking results by evaluating both the speed and effectiveness of various algorithms. This model will set a threshold of RMSD, typically 2 Å, to determine if a docking result is considered effective. It will then predict whether each algorithm can meet this threshold and how long it will take, allowing the selection of the most time-efficient algorithm without sacrificing accuracy. Together, these models provide a comprehensive approach to improving both the accuracy and efficiency of molecular docking, addressing the current challenges faced in the field.

Our contributions can be summarized as follows.

1. There is limited research on AS for molecular docking. Our proposed system, GNNAS-Dock, is the first GNN-based AS approach considering both proteins and ligands, with distinct graph representations.
2. Among the two suggested GNNAS-Dock variants, the accuracy model achieves an average RMSD value of 1.74 Å, outperforming all the tested docking methods, including DiffDock as the best standalone algorithm with RMSD of 2.95 ÅIt also achieves a success rate of 81.8% for docking RMSD values below the 2 Å threshold and 92% below 5 Å threshold, which also demonstrate its docking accuracy. For the efficiency model, it achieves an average RMSD value of 2.75 Å, with about 79.73% of RMSD values below 2Å threshold and average processing time of 29.05 seconds per instance. The performance of efficiency model beats all the other docking algorithms in average ligand RMSD and percentage of results below 5 Å threshold. Though DiffDock is better in terms of median RMSD and percentage of results below 2 Å the efficiency model takes much less time to run compared with DiffDock which takes an average time of 37.31 seconds to complete each docking tasks. Nevertheless, it is possible to outperform all those docking methods when an AS model is built specifically targeting any of those performance criteria.

For the reminder of the paper, a discussion on molecular docking and algorithm selection is delivered through existing, relevant research in the Section 2. Then, the proposed system, GNNAS-Dock,

is introduced in Section 3. Section 4 reports the computational results and analysis following a particular experimental setup. Finally, a summary of the major findings and future research plans to refine the work are provided in Section 5.

## 2 BACKGROUND

### 2.1 MOLECULAR DOCKING METHODS.

Molecular docking can be investigated under different categories, such as blind docking, redocking, and crossdocking. Blind docking refers to the situation that the binding site of the protein is unknown, requiring the docking algorithm to explore potential binding sites across the entire surface of the protein (Hassan et al., 2017). For blind docking, various docking methods exists for predicting the binding sites and potential binding poses of ligands. All the methods follow a similar algorithmic template. It is predicting potential binding poses and then refining these predictions based on some evaluative criteria. The search techniques and performance metrics differ though. In that sense, traditional docking algorithms typically rely on search methods and scoring functions to explore the possible binding configurations of a ligand in the protein and refine them iteratively. For instance, AutoDock Vina (Trott & Olson, 2010), a well-known docking method utilizes a stochastic search methods combined with a empirical and knowledge-based scoring function to predict ligand configurations. It has some variants such as SMINA (Koes et al., 2013) and QuickVina-W (Qvina-W) (Hassan et al., 2017). SMINA extends the AutoDock Vina by adjusting the scoring function and making it customizable. Besides, QuickVina-W enhances the original algorithm by optimizing the search algorithm for faster performance. These methods are mature and have been validated in many docking tasks, which make them still useful in many situations.

Recent advancements have incorporated Deep Learning (DL) into both searching and scoring steps to enhance docking quality for blind docking. GNINA (McNutt et al., 2021) employs Convolutional Neural Networks (CNNs) to score potential poses, speeding up the evaluation process while maintaining high accuracy. DSDP (Huang et al., 2023) leverages 3D CNNs to predict docking sites, then uses search and scoring functions similar to those in AutoDock Vina for the docking process. Additionally, TankBind (Lu et al., 2022) utilizes geometric deep learning to directly predict potential binding poses of ligands. Similarly, DiffDock (Corso et al., 2023) employs a generative model to dynamically simulate and rank possible binding configurations. These deep learning combined docking methods are more efficient then traditional algorithms, but traditional methods still reach higher accuracy in some situations, especially when combined with some docking site prediction tool such as p2rank (Krivák & Hoksza, 2018).

### 2.2 ALGORITHM SELECTION.

Algorithm Selection (AS) has been commonly devised as performance prediction models, mapping a set of hand-picked features representing the target problem instance to the performance of the algorithms at hand, $\Phi : \mathbf{X} \in \mathbb{R}^d \mapsto \mathcal{P}_\mathcal{A} \in \mathbb{R}^m$. Following those traditional machine learning based AS models, deep learning has also been used, mostly to eliminate the need of those explicit problem instance features. As an initial attempt, a Convolutional Neural Network (CNN) has been utilized to build an AS model for solving the Boolean Satisfiability (SAT) problem Loreggia et al. (2016).The problem instances are transformed into images by representing each character, simply appearing in the instance files, by ASCII values representing the scales of grey. The images are given to a CNN architecture to choose an algorithm for solving each instance. In Kostovska et al. (2023), transformer architecture was used for selection algorithms targeting well-known black-box optimization (BBOB) problems. In drug discovery, some meta-learning models (Olier et al., 2018; Schlender et al., 2023) are used for AS on quantitative structure activity relationships (QSAR). For AS targeting docking, in the very first study (Chen et al., 2023), ALORS (Mısır & Sebag, 2017), an algorithm recommendation system based on collaborative filtering, was accommodated to determine the optimal search box size in AutoDock 4.2 for docking tasks involving 1428 ligands on the Human Angiotensin-Converting Enzyme (ACE). The results show that ALORS outperforms all the individual algorithm setups. Following that, we earlier designed more of a traditional AS model to select the optimal docking algorithm among 6 different options for blind docking tasks with protein-ligand complexes from the PDBBind dataset (Liu et al., 2017), which contain more

than 19,000 pairs for blind docking. That work as a proof of concept demonstrated the effectiveness of AS in blind docking.

# 3 METHOD: GNNAS-DOCK

The present study introduces a Graph Neural Networks (GNN) based Algorithm Selection (AS) system for Molecular Docking (GNNAS-Dock). The goal is to automatically identify the right docking algorithm from a candidate set for a given protein-ligand pair to deliver high quality and robust blind docking. The process starts with the construction of distinct graphs representing ligands, $\mathcal{G}_L = (V_L, E_L)$, and proteins, $\mathcal{G}_P = (V_P, E_P)$, due to the significant differences in their sizes and structural complexities. Here, $V$ and $E$ denote vertices and edges, respectively.

Then, a predictive model leveraging stacking learning (Mohammed & Kora, 2023) for AS is built. In the context of our research, stacking learning involves using the output features from the first-level GNNs as inputs to a second-level meta-model for prediction. Firstly, two GNNs are built to process protein and ligand graphs separately to extract features learned from their structures. Given the distinct structural complexities of proteins and ligands, it is crucial to design specific GNN architectures for each type of graph. These architectures are specialized to efficiently process and learn from the distinct topological and chemical properties in each graph type. After extracting learned features from the individual GNNs for both proteins and ligands, the next step involves integrating these features through direct concatenation. The concatenated features serve as input to a series of dense layers which construct the second-level meta-model in the stacking learning model. This meta-model makes final predictions for assessing the performance of each docking algorithms, predicting not only the most accurate algorithm in terms of RMSD but also estimating the computational efficiency for each algorithm.

In the final configuration of GNNAS-Dock, two distinct AS models are built referring to different criteria. The first model prioritizes accuracy, which recommends the docking algorithm achieving the lowest docking quality, which is RMSD. The overall working structure of the accuracy oriented GNNAS-Dock is illustrated in Figure 1. The second model focuses more on efficiency, which chooses the algorithm that completes the docking process fastest while achieving an acceptable binding result for a given docking scenario. This dual-model approach allows users to select the optimal algorithm based on the specific requirements of their molecular docking tasks.

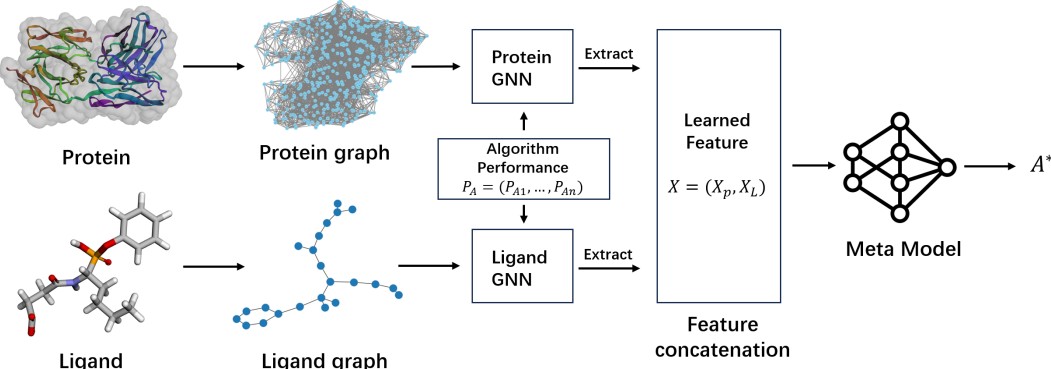

Figure 1: How accuracy oriented GNNAS-Dock operates for choosing an (near-)optimal docking algorithm for a sample protein of PDB:1a0q along with a particular ligand.

## 3.1 LIGAND GRAPH

The process of representing ligands in graphs begins by transforming the SMILES format of molecules into 3-dimensional molecular structures using the RDKit package in Python. Following this transformation, the graph representation of molecule ligand is defined as $\mathcal{G}_L = (V_L, E_L)$. In this graph, atoms are treated as nodes $v_i \in V_L$ and chemical bonds as edges $e_{ij} \in E_L$ between node $v_i$ and $v_j$.

Each node $v_i$ is associated with a set of features, which are represented in a feature matrix $X \in \mathbb{R}^{n \times D}$, i.e. $n$ denotes the number of nodes and $D$ represents the number of features for each node. In ligand molecules, several atomic attributes are used to represent the node features for $x_i \in X$, including: the 3D coordinates of atom $\vec{c}_i = (x_{ci}, y_{ci}, z_{ci})$, a one-hot encoded vector $\vec{t}_i$ for atom type, and some chemical and physical properties of ligands such as molecular weight and total valence. We represent these properties in a vector $\vec{P}$ From the perspective of the atomic structure of molecules, the atom types in a molecule can be one of the following {'C', 'H', 'O', 'N', 'F', 'P', 'S', 'Cl', 'Br', 'I', 'B'}. These features represent the feature vector $x_i = [x_{ci}, y_{ci}, z_{ci}, onehot(\vec{t}_i), \vec{P}_i]$ for each node $v_i$ and form rows in the feature matrix $X \in \mathbb{R}^{n \times 25}$. Edge features are used as edge weights in the GNNs. For edge features in our graph representation, the bond type is used since it represents how atoms are connected. Specifically, we categorize bond types into four distinct classes: single, double, triple, and aromatic. Besides these basic settings for nodes and edges, an adjacency matrix $A \in \mathbb{R}^{n \times n}$ is maintained. $A_{ij} = 1$ if node $v_i$ is connected with $v_j$ or 0 otherwise. Since the graphs are undirected, $A$ is symmetric, i.e., $A_{ij} = A_{ji}$.

After setting the overall graph for the ligand molecules, we need to design specific GNN architecture to learn the graph and output new features we want to represent the ligand. Due to the simplicity of the overall structure of ligand, simpler network could be used. Here, we mainly use three distinct GNN architectures to learn the overall features of ligands, including Graph Convolutional Network (GCN) (Kipf & Welling, 2017), Graph Attention Network (GAT) (Veličković et al., 2018), Graph Isomorphism Network (GINE) (Hu et al., 2020). The GCN aggregates neighbor information to update node features iteratively. The updating rule is: $X^{(l+1)} = \sigma\left(\tilde{D}^{-\frac{1}{2}}\tilde{A}\tilde{D}^{-\frac{1}{2}}X^{(l)}W^{(l)}\right)$ in which $\tilde{D}$ denotes the diagonal degree matrix of $\tilde{A}$, $\tilde{A} = A + I_n$ and $W^{(l)}$ denotes the weight matrix at layer $l$. This formula shows how the feature matrix $X^{(l)}$ at layer $l$ is transformed to $X^{(l+1)}$ in the next layer. For the choice of activation function $\sigma$, we decide to use ReLU. $\tilde{D}^{-\frac{1}{2}}\tilde{A}\tilde{D}^{-\frac{1}{2}}$ represents a symmetric normalization for adjacency matrix, which ensure the balance between influences of both highly connected and less connected nodes in the molecular graph. The GAT combine the graph neural network with attention mechanism and it updates the graph feature for each node $v_i$ with the formula $x_i^{(l+1)} = \big\|_{k=1}^{K} x_i^{(l+1,k)} = \sigma\left(\sum_{j \in \mathcal{N}(i) \cup \{i\}} \alpha_{ij}^{(k)} W^{(k)} x_j^{(l)}\right)$ in which $K$ represents the number of multi-head-attentions or heads, $\alpha_{ij}^{(k)}$ denotes the attention coefficients that determine the weights of node $v_j$'s feature $x_j$ when updating the feature $x_i$ for node $v_i$ in head $k \in K$. Each attention head $k$ focus on different relationships between nodes in the graph, and update the features according to it. The learned features will be concatenated, so the transformation will convert the input feature matrix from $\mathbb{R}^{n \times 25}$ to $\mathbb{R}^{n \times (K \times m)}$ where m is the output dimension in the last layer for each node. By connecting the attention mechanism, the GAT could learn different features from ligand structures. Finally, for the GINE, it updates the feature matrix with: $x_i^{(l+1)} = \text{MLP}^{(l)}\left((1 + \epsilon) \cdot x_i^{(l)} + \sum_{j \in \mathcal{N}(i)} \text{ReLU}(e_{ij}^{(l)} + x_j^{(l)})\right)$, in which $e_{ij}$ is the edge feature or weight. The transformation is combined with a multilayer perceptron with a single hidden layer, i.e., $\text{MLP}^{(l)} = \text{Sequential}(\text{Linear}(), \text{ReLU}(), \text{Linear}())$. The MLP provides an advantage over simpler transformation functions due to its layered structure, which allows for a more complex and refined processing of molecular features. The final output of the ligand GNN is a learned feature matrix $X_l \in \mathbb{R}^{n \times k}$ where $k$ is the number of algorithms for selection.

## 3.2 PROTEIN GRAPH

In this section, we will explore how proteins are converted into graphs and processed by GNNs. Initially, proteins are imported in 3D molecular structures from the Protein Data Bank (PDB) format, utilizing the PyMOL Python package, Once the protein structures are loaded, we proceed to construct the graph representation $\mathcal{G}_P = (V_P, E_P)$ for proteins.

Compared with small-molecule ligands, proteins consist of a significantly larger number of atoms and bonds. Representing them at the atomic level as ligand graphs poses computational challenges for the overall model. To reduce the computational complexity of processing protein graphs, we simplify the graph representation by using amino acids, clusters of atoms with distinct functions, to represent each node $v_i \in V_P$. The edges $e_{ij}$ in the graph are represented by peptide bonds, which connect amino acids $v_i$ and $v_j$ in the protein's structure.

For the feature matrix of nodes $X$, the feature for each node $x_i$ consists of the coordinates of centers $\vec{c}_i = (x_{ci}, y_{ci}, z_{ci})$, a one-hot encoded vector $\vec{At}_i$ identifying the amino acid from the 20 standard amino acids available in proteins, the molecular weight of the amino acid $w_i$, binary factors $HA_i$ and $HD_i$ to decide whether the amino acid is a hydrogen acceptor or donor, and the meiler embeddings $\vec{m}_i$ including 7 distinct physicochemical features (Meiler et al., 2001) for amino acids. These features collectively form the feature vector $x_i = [x_{ci}, y_{ci}, z_{ci}, onehot(\vec{At}_i), w_i, HA_i, HD_i, \vec{m}_i] \in \mathbb{R}^{32}$ for each node $v_i$. For the edges, we simply use the length of peptide bonds connecting nodes to represent edge features. In addition, for the adjacency matrix $A$ in protein, we define $A_{ij} = 1$ if peptide exists between node $v_i$ and $v_j$ or 0 otherwise. Similarly, the protein graph is undirected, so $A$ is symmetric, i.e., $A_{ij} = A_{ji}$.

The GNN architecture we applied to process the protein graph derived from the graphLambda (Mqawass & Popov, 2024), which used a combination of GCN, GAT and GINE to learn the protein graph features. As mentioned in ligand graph, different GNN layers have distinct advantages when dealing with graph data. For proteins, it is important to understand both the overall structure at the amino acid level and the detailed structure within each amino acid. Therefore, a sophisticated GNN architecture is essential for a comprehensive analysis for protein graphs. Therefore, different combination of GNNs should be experimented to determine an effective GNN architecture for processing protein graph data. Similar to the final output from protein graph, the output from ligand graph is represented as $X_p \in \mathbb{R}^{n \times k}$ where k is the number of algorithms.

### 3.3 ACCURACY MODEL

The accuracy model within the GNNAS-Dock system integrates Graph Neural Networks (GNNs) for both proteins and ligands into a combined training framework to predict the most accurate docking algorithm for each docking instance in terms of RMSD. The architecture includes a Protein GNN and a Ligand GNN, which simultaneously process graph representations of protein $\mathcal{G}_P$ and ligand $\mathcal{G}_L$ respectively. The specific structures of these GNNs have been discussed in the previous sections. They will extract some important features from the graphs automatically to represent the 3D structures of protein and ligands without defining manually. Then, a stacking ensemble learning model is strategically designed to integrate the learned results from both the Protein GNN and Ligand GNN. This model consolidates the distinct features extracted from each GNN into a unified feature set, denoted as $X_{combined} = (X_{protein}, X_{ligand})$. These combined features are then employed to train a meta-model $M$ to predict the performance of various docking algorithms based on the integrated data from both the protein and ligand models. The overall learning process is conducted end-to-end, ensuring that no manual intervention is required for tasks such as feature extraction, weight assignment, or other parameter adjustments. For the meta-model, a comparatively simple architecture is adopted, consisting of only one linear hidden layer. The input layer in the meta-model is responsible for receiving the combined feature set that concatenates the outputs of the Protein GNN and Ligand GNN, which have a dimension of $2 \times k$ for $k$ representing the feature dimension of extracted from protein and ligand models. This linear transformation of the hidden layer expands the feature dimensions to $4 \times k$ by $h(x) = W_1 x + b_1$, in which $W_1 \in \mathbb{R}^{(4 \times k) \times (2 \times k)}$ represents the weight matrix, and $b_1 \in \mathbb{R}^{4 \times k}$ is a bias vector. Then, the output layer will use the expanded features to predict the final results by $output = W_2 h(x) + b_2$. The final results $output \in \mathbb{R}^k$ show the predicted performance of each algorithm. Based on the output, we could rank the performance of each algorithm, and return the best one for solving the docking problem. The specific process of Accuracy Model in GNNAS-Dock could be reviewed in Algorithm 1.

---

**Algorithm 1** GNNAS-Dock Accuracy Model Generation and Prediction

---

**Input:** Protein and Ligand graphs for training $\mathcal{G}_L, \mathcal{G}_P$; performance of different docking algorithms in terms of a particular metric, $Y \in \mathbb{R}^{n \times k}$; a single Protein and Ligand pair for docking, $\mathcal{G}_{L_{new}}, \mathcal{G}_{P_{new}}$; $n$ is the total number docking tasks; $k$ represents the number of docking algorithms for selection.

  1: $\Phi_Y : \{\mathcal{G}_L, \mathcal{G}_P\} \mapsto Y$                                  // performance prediction model
  2: $Y_{est} = (Y_{A_1}, \ldots, Y_{A_k}) \leftarrow \Phi_Y(\mathcal{G}_{L_{new}}, \mathcal{G}_{P_{new}})$
  3: $A^* = \arg\min_{A_i} Y_{est}$

---

## 3.4 EFFICIENCY MODEL

In the accuracy model, only the performance of each docking algorithm is considered. However, it is possible that some accurate models may take longer to run. Additionally, although lower RMSD is better, binding can be viewed as successful if its RMSD is below a threshold, which is usually 2 Å (Castro-Alvarez et al., 2017). Therefore, an efficiency model is also built for choosing the quickest good performing algorithm under a specific performance threshold.

Two predictive models are constructed based on the features extracted from both protein and ligand through GNNs. One model predicts whether each docking algorithm can successfully solve a docking problem, with outcomes categorized as 1 ("solvable") or 0 ("unsolvable"). If the docking algorithm could generate a result with RMSD lower than 2 Å, it is categorized as 1; otherwise, it will be classified as 0. Since this is a binary classification model, we changed the output layer in the meta-model to be $output = \sigma(W_2 h(x) + b_2)$ in which $\sigma$ denotes the sigmoid function. It will return the probabilities of whether the docking methods could solve the docking problem. Then, the other model is built to predict the time required for each docking algorithm to produce its result, which uses the same GNN structure as the accuracy model to extract features from protein and ligand graphs, and same meta-model structures. These two models are trained separately since their objectives are different. For each docking task to be handled, only those algorithms predicted to solve the problem, i.e., categorized as 1, are considered. Among them, the algorithm with the shortest predicted time is recommended. When there is no algorithm predicted to meet the RMSD threshold of 2 Å the algorithm predicted to be the fastest is chosen. This methodology provides a balanced selection concerning docking quality and speed. The complete procedures are shown in Algorithm 2.

---

**Algorithm 2** GNNAS-Dock Efficiency Model Generation and Prediction

---

**Input:** Protein and Ligand graphs for training $\mathcal{G}_L, \mathcal{G}_P$; the spent execution times for the docking algorithms $T \in \mathbb{R}^{n \times k}$; binary success outcomes for docking algorithms $S \in \mathbb{R}^{n \times k}$; a Protein and Ligand pair for docking $\mathcal{G}_{L_{new}}, \mathcal{G}_{P_{new}}$; $n$ is the total number docking tasks; $k$ represents the number of candidate docking algorithms for selection.

1: $\Phi_T : \{\mathcal{G}_L, \mathcal{G}_P\} \mapsto T$                                         // runtime prediction model
2: $\Phi_S : \{\mathcal{G}_L, \mathcal{G}_P\} \mapsto S$                                     // binary success prediction model
3: $T_{est} = (T_{A_1}, \ldots, T_{A_k}) \leftarrow \Phi_T(\mathcal{G}_{L_{new}}, \mathcal{G}_{P_{new}})$
4: $S_{est} = (S_{A_1}, \ldots, S_{A_k}) \leftarrow \Phi_S(\mathcal{G}_{L_{new}}, \mathcal{G}_{P_{new}})$
5: **if** $S_{A_j} = 1, \exists j$ **then**
6:     $A^* = \arg\min_{A_j} \{T_{A_j}, \forall j \text{ where } S_{A_j} = 1\}$
7: **else**
8:     $A^* = \arg\min_{A_i} T_{est}$

---

## 4 EXPERIMENTS AND RESULTS

In this study, we evaluate the performance of the GNNAS-Dock system using a selection of nine docking algorithms. These algorithms include DiffDock, TankBind, DSDP, GNINA, SMINA, Qvina, and combinations that integrate docking site prediction tool, such as GNINA+p2rank, SMINA+p2rank, and Qvina+p2rank. This portfolio of algorithms is diverse since it contains both traditional docking algorithms and some advanced methods integrating deep learning techniques. These algorithms are all state-of-the-art docking algorithms in their own types.

The dataset employed for evaluating the algorithms is derived from the refined set of the PDBBind database (Liu et al., 2017), which consists of approximately 5,300 pairs of protein-ligand complexes. This dataset is highly regarded within the drug discovery community for its quality and relevance, making it an ideal choice for rigorous testing of docking algorithms. For constructing graphs for proteins and ligands, we utilize the Python package Graphein (Jamasb et al., 2022).

For measuring docking performances, symmetry-corrected root mean square deviation (RMSD) is used as the metric (Meli & Biggin, 2020), as in the recent docking approaches such as DiffDock. It additionally takes molecular symmetry into count, providing a more accurate evaluation.

All experiments were conducted through the implementations in Python, on a server equipped with 2 AMD EPYC 7763 64-Core CPUs, 8 NVIDIA RTX 3090 GPUs, and 256 Gigabytes of RAM, running Ubuntu OS version 20.04.6.

## 4.1 PROTEIN AND LIGAND GRAPH ARCHITECTURES

To refine the predictive capabilities of GNNAS-Dock, various combinations of graph neural network (GNN) architectures for protein and ligand graphs are tested. As mentioned in the previous part, 3 GNN architectures are used for exploring ligand features, including GCN, GAT and GINE. For protein, we use the combination of GCN, GAT and GINE as the graphLamda uses different node features for protein graphs. We split the dataset into a training and testing set with a ratio of 0.3. Solely for having a general view on the success of each combination, the learning rate of 0.0001 with 50 epochs is used. The best architecture combination is determined based on their prediction qualities, as reported in Table 1. Each value denotes the average docking performance in RMSD for a specific pair of protein and ligand network architectures on the test set.

Table 1: Average docking performance in RMSD for each combination of protein (column) and ligand (row) graph architectures.

|  | GCN_GAT_GINE | GCN_GAT | GCN_GINE | GAT_GINE | GCN | GAT | GINE |
|---|---|---|---|---|---|---|---|
| **GCN** | 1.879 | 2.015 | 1.862 | 1.849 | 2.238 | 1.941 | 1.908 |
| **GAT** | **1.833** | 2.108 | 1.970 | 1.888 | 2.311 | 1.984 | 2.007 |
| **GINE** | 1.883 | 2.198 | 1.858 | 1.903 | 2.300 | 1.977 | 1.940 |

It is evident that the combination of GCN_GAT_GINE for the protein graph and GAT for the ligand graph achieves the lowest RMSD value, indicating the highest docking accuracy among the tested configurations. Consequently, this specific pairing has been selected for building the GNNAS-Dock. Additionally, it should be noted that any of these pairs outperform the overall, single best docking algorithm of DiffDock, which is further discussed below.

## 4.2 COMPUTATIONAL RESULTS

The results of all the individual docking algorithms and GNNAS-Dock models on the PDBBind dataset are reported in Table 2. When predicting potential binding poses for ligands in the protein, each docking algorithm will generate multiple candidates for choosing. In this study, only the best-docked result from each algorithm for each docking task will be selected for analysis.

Table 2: Descriptive statistics in terms of RMSD combined with average running time for different docking algorithms in the test set.

| Algorithm | RMSD | | % Below Threshold | | |
|---|---|---|---|---|---|
| | **Mean** | **Median** | **2 Å** | **5 Å** | **Avg. Time** |
| DiffDock | 2.95 | **0.65** | **84.36%** | 90.83% | 37.31 |
| DSDP | 5.23 | 1.97 | 53.41% | 71.26% | 0.82 |
| GNINA | 9.21 | 2.40 | 46.58% | 59.82% | 22.11 |
| GNINA + p2rank | 3.40 | 1.77 | 53.55% | 74.29% | 169.76 |
| QVina | 9.00 | 4.41 | 36.32% | 53.00% | 13.92 |
| QVina + p2rank | 3.88 | 2.47 | 45.97% | 68.30% | 427.65 |
| Smina | 9.26 | 4.84 | 34.67% | 51.07% | 19.60 |
| Smina + p2rank | 3.90 | 2.63 | 44.80% | 67.61% | 171.84 |
| TankBind | 3.64 | 1.71 | 59.13% | 89.04% | **0.03** |
| GNNAS-Dock (Accuracy) | **1.74** | 0.71 | 81.80% | **92.00%** | 66.85 |
| GNNAS-Dock (Efficiency) | 2.75 | 0.81 | 79.73% | 91.38% | 29.05 |
| Oracle | 0.67 | 0.47 | 96.07% | 99.65 | 79.80 |

From the performance comparison, the Accuracy Model stands out in enhancing docking accuracy across various metrics. It achieves the lowest mean RMSD values at 1.74 Å, which is much lower

than each single docking algorithm. Notably, DiffDock presents a stronger median RMSD of 0.65 Å, slightly better than the Accuracy Model. In terms of percentage predictions below specific RMSD thresholds, the Accuracy Model records 81.80% below the 2 Å threshold and a remarkable 92.00% below the 5 Å threshold, which shows a much greater performance than most of the docking algorithms. Though DiffDock slightly outperforms the Accuracy Model at the 2 Å threshold, the Accuracy Model dominates at the 5 Å threshold. These results suggest that the Accuracy Model could effectively eliminate instances of extremely poor docking outcomes and consistently limit the results to a more acceptable range.

Despite the success of GNNAS-Dock (Accuracy), there is a substantial variance in performance and efficiency among the algorithms. The accuracy model, for instance, averages a runtime of 66.85 seconds per docking task. This is comparatively long, especially when aligned against some of the faster alternatives. DiffDock as the best standalone algorithm performs inferior to GNNAS-Dock (Accuracy), it is relatively fast, requiring 37.31 seconds on average. There are much faster algorithms such as TankBind and DSDP, yet they offer lower docking quality though. TankBind stands out remarkably, completing dockings in approximately 0.03 seconds. Given these observations, there's a challenge for docking algorithms to balance performance with efficiency. To address this, GNNAS-Dock (Efficieny) is offered for both considering docking quality and runtime requirements at the same time. From the Table 2, GNNAS-Dock (Efficieny), indeed, delivers a well-balanced behaviour between speed and accuracy. It achieves a mean RMSD of 2.75 Å and a median of 0.81 Å, with 79.73% of its dockings below the 2 Å threshold and 91.38% below 5 Å. These performances all indicate reasonably accurate docking predictions that are obviously better than most of the other single docking algorithms. Compared with DiffDock, the single docking algorithm with best performance, the efficiency model has a lower mean RMSD and higher percentage of results within the 5 Åthreshold. More importantly, it operates with an average time of 29.05 seconds per docking, significantly faster than most of the other docking algorithms including DiffDock. Based on our setting of using 2 Å as the threshold for deciding the success of a docking problem, the efficiency model reaches a compelling balance between docking accuracy and running efficiency. This makes it an attractive option in high-throughput scenarios where quick and accurate results are more important.

## 5 CONCLUSIONS

This study presents a novel and robust approach called a Graph Neural networks (GNN) based Algorithm Selection (AS) system for molecular docking (GNNAS-Dock). In this system, we design two distinct models for recommending suitable docking algorithms with different criterion, i.e., accuracy and efficiency. Through extensive experimentation on the PDBBind refined set, both models demonstrate significant improvements over single docking algorithms for blind docking tasks in terms of accuracy. The accuracy model reaches a much lower mean RMSD than all the other docking algorithms, indicating its superior precision in predicting the most accurate docking algorithm for docking tasks. The efficiency model achieves a comparatively good performance for docking tasks in less than half the running time of the accuracy model, showing a great balance between accuracy and efficiency.

While GNNAS-Dock system shows significant potential in enhancing docking efficiency and accuracy, the built models can still be improved in terms of generalization. The overall experiment is conducted with the PDBBind refined set, which may not fully represent the diverse range of molecular interactions encountered in pharmaceutical research. To address this, future development could focus on expanding dataset diversity by including more protein-ligand pairs, which would improve model robustness and real-world applicability. Additionally, structural optimizations including the adjustments for GNN architectures, could be applied to enhance GNNAS-Dock further.

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
