# OpenReview forum: "GNNAS-Dock: Budget Aware Algorithm Selection with Graph Neural Networks for Molecular Docking"
_ICLR.cc/2025/Conference — Submitted to ICLR 2025_

### Official Review · Reviewer_bQKw · 2024-11-01

**Soundness:** 2
**Presentation:** 3
**Contribution:** 2
**Rating:** 3
**Confidence:** 3

**Summary:**

This paper introduces GNNAS-Dock, an algorithm selection method tailored for molecular docking. GNNAS-Dock first generates molecular representations for both ligand and protein by combining GCN, GAT, and GIN models. These representations are concatenated and then passed through two separate models—an accuracy model and an efficiency model—that predict the performance of each algorithm. Experimental results demonstrate GNNAS-Dock’s performance compared to previous molecular docking approaches.

**Strengths:**

- The paper is easy to follow.
- The use of two criteria, accuracy and efficiency, for algorithm selection is a reasonable approach that adds value.

**Weaknesses:**

- Is the Accuracy Model and Efficiency Model trained using supervised learning with a labeled dataset that includes labels for both accuracy and efficiency?
- When a new molecular docking algorithm is released, does GNNAS-Dock need to be retrained from scratch? Would this require a new labeled dataset for the accuracy and efficiency of the new docking algorithms? Could transfer learning be applied to GNNAS-Dock in this scenario? If transfer learning is not feasible, this would represent a significant weakness.
- Table 2 shows that DiffDock achieves comparable or even superior performance to GNNAS-Dock. Given that GNNAS-Dock requires additional overhead compared to DiffDock, it raises the question: why would practitioners choose GNNAS-Dock over DiffDock? Would DiffDock be a preferable choice for practical applications?
- Is there any baseline in the literature for algorithm selection that could be used as a comparison, rather than individual molecular docking algorithms?
- I suggest moving the explanations of GCN, GAT, and GIN from the Methods section to a Preliminary section. This would allow the Methods section to focus solely on the novel methodological contributions, which would better clarify the paper’s contributions.
- Typo: Line 221 "From"

**Questions:**

See Weaknesses Section.

---

> ### Author Response · Authors · 2024-11-23
>
> Thank you for your valuable opinions. We appreciate some suggestions you've provided and are considering incorporating them into our revisions. To address the weaknesses you pointed out, we offer the following responses:
>
> 1.Training of Accuracy and Efficiency Models: The models for assessing accuracy and efficiency in GNNAS-Dock are indeed trained using a labeled dataset, but labels such as accuracy are derived from RMSD values, and efficiency is based on the time required for running the docking process. The models will predict these metrics for different docking algorithms based on the input features of the molecular pairs, and make selection according to the predicted results.
>
> 2.Updating GNNAS-Dock with New Algorithms: GNNAS-Dock is designed to accommodate new molecular docking algorithms without requiring retraining from scratch. For a new algorithm, we can train an additional model specifically for that algorithm using the docking results in the same dataset and same metrics. This model can then be integrated with existing models to enhance the algorithm selection process. Transfer learning is certainly feasible and beneficial, particularly when new algorithms share similarities with those already in the dataset, such as between SMINA and QVinaW, since they are both built based on AutoDock Vina.
>
> 3.Comparison with DiffDock: While DiffDock shows strong performance on the PDBBind dataset, which it was trained on, GNNAS-Dock is designed to offer flexibility across various datasets and scenarios. For example, in the DUDE dataset, we find that DiffDock does not perform better than other docking algorithms, but GNNAS-Dock could still reach a better performance than other docking algorithms with more stability. This is shown in Table 1, where we run different docking methods and GNNAS-Dock with part of DUDE dataset [1] and evaluate them using AA-Score [2]. We find that GNNAS-Dock performs better than each single docking method in terms of mean, median and threshold, and it has lower variance, which means the method reaches a more stable result. GNNAS-Dock's value lies in its ability to adapt and select among multiple docking algorithms based on specific molecular contexts, providing an approach that can complement each single methods' strengths.
>
> | Algorithm | Mean         | Std         | Median         | 0AA          | 5AA          |
> |-----------|--------------|-------------|-------------|--------------|--------------|
> | DiffDock  | 1.58317e+06  | 6.843969e+07| -6.14      | 75.384615 | 79.538462    |
> | GNINA     | 2.431891e+06 | 8.094182e+07| -7.82      | 96.102564 | 96.102564    |
> | SMINA     | 7.221649e+05 | 1.313745e+07| -9.07      | 97.179487 | 97.179487    |
> | TankBind  | 2.187797e+07 | 8.707069e+08| -1.19     | 51.384615 | 56.769231    |
> | DSDP      | 1.396005e+06 | 3.248848e+07| -9.07      | 83.897436 | 83.948718    |
> | GNNAS   | -3.988457e+00| 1.230933e+02| -9.22      | 98.256410 | 98.769231    |
> | best      | -9.578611e+00| 3.106873e+00| -9.66      | 99.179487 | 99.435897    |
>
> **Table 1:** AS model with DUDE dataset.
>
> 4.Baseline Comparisons: Given the specialized nature of algorithm selection in molecular docking, there isn’t a universal baseline model that applies across all contexts. Each problem within this domain might require a distinct algorithm selection model tailored to its specific requirements and constraints. For instance, in the previous study of algorithm in molecular docking, researchers manually extract the 2D features of protein-ligand pairs and use traditional regression model for selection. However, in our study, we use graph representation with GNN to make selections. Therefore, it is hard to find a baseline for comparison.
>
> Reference:
>
> [1] Mysinger MM, Carchia M, Irwin JJ, Shoichet BK J. Med. Chem., 2012, Jul 5. doi 10.1021/jm300687e.
>
> [2] Pan, Xiaolin, et al. “AA-Score: A New Scoring Function Based on Amino Acid-Specific Interaction for Molecular Docking.” Journal of Chemical Information and Modeling, vol. 62, no. 10, 22 Apr. 2022, pp. 2499–2509, https://doi.org/10.1021/acs.jcim.1c01537. Accessed 22 Nov. 2024.

---

> > ### Comment · Reviewer_toF4 · 2024-11-23
> >
> > 3.  Evaluating on DUDE is a good choice, but virtual screening metrics need to be reported. AA-score is largely irrelevant in this setting.
> > 4. Examples of simple base lines would be min-rank, ave-rank, etc. Something more complex might be belief theory (https://pubs.acs.org/doi/abs/10.1021/ci7004498).
> >
> > Hopefully the authors can devise robust, meaningful evaluation training and evaluation frameworks for a future submission.

---

> > > ### Author Response · Authors · 2024-11-25
> > >
> > > Thank you for your comment. We want to clarify your misunderstandings.
> > > 1. Our study utilizes the DUDE dataset primarily for blind docking purposes, not virtual screening. We used blind docking to generate and assess potential docking poses for protein and active compounds in the DUDE dataset. In this situation, scoring function such as AA-Score indeed matters. We understand there are better datasets for performing the experiments, but since we were just running some other experiments with DUDE dataset, we could quickly access and use the existing data for our experiments with the GNNAS-Dock model. In addition, some other works also use DUDE to perform such blind docking instead of virtual screening, like EDock [1]. We want to show that the GNNAS-Dock could also adopt other datasets and evaluating metrics instead of just PDBBind and RMSD.
> > > 2. Our model is specifically designed to be evaluated against state-of-the-art docking methods, not traditional baselines like min-rank or ave-rank. These baselines ultimately reduce to selecting a single best-performing method, which does not demonstrate the adaptive algorithm selection our model employs. Therefore, it is unnecessary to compare with these baselines. By outperforming each individual state-of-the-art method, our model proves its utility in dynamically choosing the optimal docking strategy which is a capability that static baselines do not have. However, we do have an oracle baseline, which represents the situation that the model could always pick the optimal docking method for each protein-ligand pairs, so it should have the best performance. We want to see if our model is close to the most optimal situation to see if there is still space for improvement.
> > >
> > > References:
> > > [1] Zhang, W., Bell, E.W., Yin, M. et al. EDock: blind protein–ligand docking by replica-exchange monte carlo simulation. J Cheminform 12, 37 (2020). https://doi.org/10.1186/s13321-020-00440-9

---

### Official Review · Reviewer_WFX7 · 2024-11-02

**Soundness:** 2
**Presentation:** 2
**Contribution:** 2
**Rating:** 3
**Confidence:** 3

**Summary:**

This paper introduces GNNAS-Dock, a novel Graph Neural Network-based algorithm selection system for molecular docking. The system aims to address two key challenges: (1) selecting the most accurate docking algorithm in terms of RMSD, and (2) maintaining computationally efficiency and acceptable accuracy. The approach was validated on the PDBBind 2020 refined dataset across 9 different docking algorithms, demonstrating improvements accuracy compared to individual algorithms while maintaining efficiency.

**Strengths:**

1. The authors conducted a comprehensive comparison experiment with the individual method, demonstrating potential improvements in accuracy while maintaining computational efficiency.
2. The authors innovatively address an important problem by applying stacking learning to molecular docking, where ensemble learning and algorithm selection approaches have been previously overlooked.

**Weaknesses:**

1. Some recently published SOTA works are missing (according to the ICLR reviewer guide, these works do not qualify as concurrent work), e.g. DiffDock-L.
2. The paper lacks comparison with other algorithm selection methods. While the comparison with individual algorithms demonstrates the effectiveness of ensemble learning or algorithm selection (which is an expected outcome), it does not sufficiently validate the merits of the authors' proposed core methodology.

**Questions:**

1. Given the recent advances in molecular docking, particularly the emergence of improved variants like DiffDock-L (which has shown promising results in blind docking scenarios), would the authors consider expanding their comparative analysis to include such state-of-the-art methods?
2. Would you consider adding comparative experiments with other algorithm selection methods cited in your related work? This could help better demonstrate the specific advantages of your proposed approach beyond the expected benefits of ensemble learning.
3. The results presented in Table 2 appear to be inconsistent with previously reported results in the literature, particularly for DiffDock. Notably, the success rate for RMSD<2Å is significantly higher than previously reported values. Could the authors provide more detailed information about their experimental settings to clarify these discrepancies?

---

> ### Author Response · Authors · 2024-11-23
>
> Thank you for your review and comments. Below are our responses to the specific issues you raised:
>
> 1.DiffDock-L Evaluation: We tested DiffDock-L in the PDBBind whole refined set containing 5,316 protein-ligand pairs , and found it did not perform better than normal DiffDock in Table 1.
>
> | Algorithm | Mean    | Std      | 50%     | 2A      | 5A      |
> |-----------|---------|----------|---------|---------|---------|
> | diff-L    | 3.25351 | 8.143872 | 0.725120| 80.570099| 89.218538|
> | diff      | 2.96568 | 7.666727 | 0.660356| 83.866589| 90.498352|
>
> **Table 1:** Comparative Results of DiffDock-L and DiffDock.
>
> We want to emphasize that our study presents a flexible framework rather than a fixed model. This flexibility allows for the inclusion of various docking methods, such as DiffDock-L, and the adaptation of other evaluation metrics such as some scoring functions to enrich our algorithm selection portfolio. Our approach aims to leverage the complementary strengths of existing docking techniques to enhance both the accuracy and efficiency of molecular docking.
>
> 2.Comparison with Other Algorithm Selection Methods: There is really limited study for the algorithm selection in molecular docking, and our work is the first one using GNN for AS in the context of molecular docking. Due to the scarcity of similar studies, direct comparisons are challenging. However, we acknowledge the importance of evaluating our methodology against other established algorithm selection methods and will consider building some other algorithm selection models for molecular docking for future research.
>
> 3.DiffDock’s Discrepency:  The discrepancies noted in our results versus original DiffDock outcomes arise from our dataset choice. The original DiffDock is trained in a large subset of PDBBind dataset and only tested in a subset of 363 protein-ligand pairs, but we validated our Algorithm Selection model on a significantly larger validation set of 1451 pairs derived from PDBBind refined set.  Among the 1451 pairs, some have been used for training the DiffDock, so it is unsurprising that DiffDock performs very well in the dataset. However, when we use the DiffDock in other datasets, such as DUDE dataset [1]. We find that DiffDock does not perform better than other docking methods in terms of AA-Score function, which is shown in Table 2. For the AA-Score [2], the lower the value, the stronger the connection between docked results and receptors. From the Table 2, we can see that the results of DiffDock is worse than other traditional docking methods such as GNINA and SMINA. However, our Algorithm Selection model is better than all the other single docking algorithms in terms of mean, median, a threshold of 0 and a threshold of 5.
>
> | Algorithm | Mean         | Std         | Median         | 0AA          | 5AA          |
> |-----------|--------------|-------------|-------------|--------------|--------------|
> | DiffDock  | 1.58317e+06  | 6.843969e+07| -6.14      | 75.384615 | 79.538462    |
> | GNINA     | 2.431891e+06 | 8.094182e+07| -7.82      | 96.102564 | 96.102564    |
> | SMINA     | 7.221649e+05 | 1.313745e+07| -9.07      | 97.179487 | 97.179487    |
> | TankBind  | 2.187797e+07 | 8.707069e+08| -1.19     | 51.384615 | 56.769231    |
> | DSDP      | 1.396005e+06 | 3.248848e+07| -9.07      | 83.897436 | 83.948718    |
> | GNNAS   | -3.988457e+00| 1.230933e+02| -9.22      | 98.256410 | 98.769231    |
> | best      | -9.578611e+00| 3.106873e+00| -9.66      | 99.179487 | 99.435897    |
>
> **Table 2:** AS model with DUDE dataset.
>
> Reference:
>
> [1] Mysinger MM, Carchia M, Irwin JJ, Shoichet BK J. Med. Chem., 2012, Jul 5. doi 10.1021/jm300687e.
>
> [2] Pan, Xiaolin, et al. “AA-Score: A New Scoring Function Based on Amino Acid-Specific Interaction for Molecular Docking.” Journal of Chemical Information and Modeling, vol. 62, no. 10, 22 Apr. 2022, pp. 2499–2509, https://doi.org/10.1021/acs.jcim.1c01537. Accessed 22 Nov. 2024.

---

### Official Review · Reviewer_toF4 · 2024-11-02

**Soundness:** 2
**Presentation:** 2
**Contribution:** 2
**Rating:** 3
**Confidence:** 5

**Summary:**

GNNAS-DOCK processes a ligand and protein using graph neural networks to predict which docking tool will provide the best result (algorithm selection).  Both the accuracy and computational time of each algorithm is predicted so they can be simultaneously optimized.

**Strengths:**

This is an interesting, practical idea for exploiting/rectifying the fact that different docking tools have different performance characteristics on different systems.

**Weaknesses:**

The evaluation is done on a random split of PDBBind, which means there is massive information leakage between the training and test sets. Given this methodological choice, it is unlikely any reported results will generalize to realistic scenarios.

Only the redocking task is considered, which is not of much practical interest.

An unbiased assessment of Table 2 is unlikely to reach the same conclusion of the authors that their tool is a better approach than using DiffDock (unsurprising as DiffDock is being evaluated on its training set).

There is no analysis of the model's predictions - it would be very interesting to learn if there is an understandable pattern to what systems different docking tools are predicted to do better on (presumably whether the system is in the method's training set, for ML methods, is highly relevant).

There are concerning methodological choices in the neural network construction.  Raw Cartesian coordinates are used as features, and not as part of an equivariant GNN.  It does not appear data augmentation was used to learn equivariance. This means the trained models are highly brittle and unlikely to generalize.  The protein graph only connects along peptide bonds, so it is a line (so why use a graph representation?).  The graph pooling operator isn't disclosed in the text.

**Questions:**

Do you augment with random translations/rotations to compensate for using Cartesian coordinates as features (or apply some other approach to induce equivariance)?

If the protein graph is only connected along peptide bonds, why not treat as sequence data?

Can you say anything about what systems different docking tools are predicted to perform well at?  What percentage of the time are different tools predicted? Is there any correlation to protein families?

How well does the method work with a more meaningful train/test split, such as provided with PLINDER?

---

> ### Author Response · Authors · 2024-11-23
>
> Thank you for your detailed feedback and insightful observations on our study. Below are our responses to address some limitations and questions you pointed.
>
> 1.Graph Construction Misexpression: We appreciate your attention to the details in our manuscript regarding the construction of protein graphs. Indeed, our model incorporates multiple connection types between residues, including peptide bonds, hydrogen bonds, disulfide bonds, and ionic bonds, not just peptide bonds as initially implied. We will revise the text to accurately reflect these details and ensure the comprehensive nature of our graph representation is clearly communicated.
>
> 2.Docking Task and Evaluation Framework: Our study aims to present a versatile framework rather than a fixed model, applicable to various docking scenarios including both redocking and blind docking. As part of our evaluation, we used subsets of the DUDE dataset [1] to perform docking using five methods (DiffDock, GNINA, SMINA, TankBind, DSDP), assessing the outcomes with the AA-Score function [2]. This approach allowed us to compare the performance of the algorithm selection model against individual docking methods.
>
> | Algorithm | Mean         | Std         | Median         | 0AA          | 5AA          |
> |-----------|--------------|-------------|-------------|--------------|--------------|
> | DiffDock  | 1.58317e+06  | 6.843969e+07| -6.14      | 75.384615 | 79.538462    |
> | GNINA     | 2.431891e+06 | 8.094182e+07| -7.82      | 96.102564 | 96.102564    |
> | SMINA     | 7.221649e+05 | 1.313745e+07| -9.07      | 97.179487 | 97.179487    |
> | TankBind  | 2.187797e+07 | 8.707069e+08| -1.19     | 51.384615 | 56.769231    |
> | DSDP      | 1.396005e+06 | 3.248848e+07| -9.07      | 83.897436 | 83.948718    |
> | GNNAS   | -3.988457e+00| 1.230933e+02| -9.22      | 98.256410 | 98.769231    |
> | best      | -9.578611e+00| 3.106873e+00| -9.66      | 99.179487 | 99.435897    |
>
> **Table 1:** AS model with DUDE dataset.
>
> For the AA-Score, the lower the value, the stronger the connection between docked results and receptors. From the Table, we can see that the results of Algorithm Selection is better than all the other single docking algorithms in terms of mean, median, a threshold of 0 and a threshold of 5, which means that the Algorithm Selection is practical in real blind docking situations. We will release the code soon.
>
> 3.Analysis of Docking Tool Performance: We acknowledge the importance of understanding the specific contexts in which different docking methods outstands. To address this, our ongoing work involves employing clustering and other analytical methods to segment the dataset into distinct groups. These groups will help identify conditions under which certain algorithms might perform optimally. This analysis aims to provide deeper insights into the strengths and weaknesses of each docking method, which could also facilitate more informed algorithm selection.
>
> 4.Concerns About Methodology and Data Handling: Regarding the use of raw Cartesian coordinates, we are exploring data augmentation with random translation. We also tested the importance of the coordinates as an attribute by removing them.
>
> | Algorithm | Mean    | Std     | Median     | 2A      | 5A      |
> |-----------|---------|---------|---------|---------|---------|
> | AS_no_coord  | 2.05863 | 4.445841| 1.522437| 80.840799| 90.075810|
> | AS_original  | 2.096518| 5.189157| 1.453644| 82.288077| 90.696072|
>
> **Table 2:** Results of removing the Cartesian coordinates in graph.
>
> In the Table 2, the AS_no_coord represents the results of our algorithm selection model after removing the Cartesian coordinates from the protein graph construction. Compared with the original AS model (AS_original), AS_no_coord shows a similar performance, which might suggest Cartesian coordinates might not be an important node feature.
>
> For the dataset choice, our algorithm selection model currently utilizes a validation set rather than a test set, primarily to demonstrate its efficacy in selecting appropriate docking methods. Recognizing the need for rigorous validation, we plan to implement cross-validation in future studies to ensure robustness. We will also consider incorporating the PLINDER into our experiments to ensure the relevance in practical drug discovery scenarios.
>
> Reference:
>
> [1] Mysinger MM, Carchia M, Irwin JJ, Shoichet BK J. Med. Chem., 2012, Jul 5. doi 10.1021/jm300687e.
>
> [2] Pan, Xiaolin, et al. “AA-Score: A New Scoring Function Based on Amino Acid-Specific Interaction for Molecular Docking.” Journal of Chemical Information and Modeling, vol. 62, no. 10, 22 Apr. 2022, pp. 2499–2509, https://doi.org/10.1021/acs.jcim.1c01537. Accessed 22 Nov. 2024.

---

### Official Review · Reviewer_Jbng · 2024-11-03

**Soundness:** 1
**Presentation:** 1
**Contribution:** 2
**Rating:** 3
**Confidence:** 5

**Summary:**

In this paper, the authors propose GNNAS-Dock, a GNN-based algorithm selection model for molecular docking that combines the best-performing docking algorithms rather than relying on a single one. GNNAS-Dock consists of two submodels: the Accuracy model, which selects the docking method with the highest performance, and the Efficiency model, which prioritizes models with faster runtimes while maintaining suitable accuracy. GNNAS-Dock’s performance is compared with single-model approaches based on the percentage of ligands achieving RMSD < 2 Å and runtime efficiency.

**Strengths:**

- The use of multiple models with GNN-based Algorithm Selection to choose the most accurate or efficient model makes GNNAS-Dock unique.
- The efficiency model demonstrates runtime improvements.
- The accuracy model shows improvement in mean RMSD.

**Originality:** GNNAS-Dock presents an innovative approach by creating a GNN-based AS model that incorporates multiple models, aiming to select them based on accuracy and efficiency.

**Weaknesses:**

- The authors highlight in the introduction and related work sections that they rely solely on the RMSD metric, which is becoming outdated in molecular docking. RMSD does not guarantee physicochemical plausibility, so I recommend they consider the PoseBuster[1] study for alternative approaches.
- Although the primary aim of the study is to select the best algorithm, the results suggest that DiffDock alone remains accurate and efficient, calling into question the effectiveness of the algorithm selection model.
- The RMSD metric used in molecular docking and Structure-Based Drug Design (SBDD) does not always yield bioactively, physically, or chemically plausible structures, as shown by studies like Posebuster[1], PoseCheck[2], PoseBench[3], and CompassDock[4]. Unfortunately, these metrics were not included in the benchmark.
- Some models may have been trained on different time splits, and if the training dataset overlaps with the validation or test sets among the chosen models, it could lead to information leakage [5]. This potential issue is insufficiently discussed.
- Details on data preparation, time-splitting, and training are inadequately explained.

**Significance:** The study does not demonstrate enough effectiveness in terms of accuracy and efficiency, which limits its applicability for scalability.

**Quality:** The paper lacks sufficient evidence and relies primarily on empirical results, without any physicochemical validation as noted above. Additionally, it is not up-to-date with the current literature, especially in the benchmarking and validation sections.

**Clarity:** The paper contains numerous grammatical errors that need revision, and the equations and annotations do not align with scientific paper standards. I recommend the authors revise the text to clarify their methods, moving away from narrative elements.

**Reproducibility:** As the code is not shared, it is currently impossible to verify whether the reported results are reproducible.


### **References**

[1] Martin Buttenschoen, Garrett M Morris, and Charlotte M Deane. Posebusters: Ai-based docking methods fail to generate physically valid poses or generalise to novel sequences. Chemical Science, 15(9):3130–3139, 2024.

[2] Charles Harris, Kieran Didi, Arian R Jamasb, Chaitanya K Joshi, Simon V Mathis, Pietro Lio, and Tom Blundell. Benchmarking generated poses: How rational is structure-based drug design with generative models? arXiv preprint arXiv:2308.07413, 2023.

[3] Alex Morehead, Nabin Giri, Jian Liu, Jianlin Cheng. Deep Learning for Protein-Ligand Docking: Are We There Yet? arXiv preprint arXiv:2405.14108, 2024.

[4] Ahmet Sarigun, Vedran Franke, Bora Uyar, Altuna Akalin. CompassDock: Comprehensive Accurate Assessment Approach for Deep Learning-Based Molecular Docking in Inference and Fine-Tuning. arXiv:2406.06841, 2024.

[5] Janani Durairaj et al. PLINDER: The protein-ligand interactions dataset and evaluation resource. bioRxiv, 2024

**Questions:**

- In the statement,
> "Given the distinct structural complexities of proteins and ligands, it is crucial to design specific GNN architectures for each type of graph,"

  why did you choose to create separate GNNs for proteins and ligands instead of using a single GNN for protein-ligand pairs?
- Why did you limit atom types to {‘C’, ‘H’, ‘O’, ‘N’, ‘F’, ‘P’, ‘S’, ‘Cl’, ‘Br’, ‘I’, ‘B’}? Was there a reason for excluding heavier atoms?
- Did you incorporate edge features in the GNN models?
- How does this differ from a coarse-grained approach?
> "To reduce the computational complexity of processing protein graphs, we simplify the graph representation by using amino acids, clusters of atoms with distinct functions, to represent each node $v_i \in V_P$."
- Did you also use an AS approach to select the GNN models?

---

> ### Author Response · Authors · 2024-11-23
>
> Thank you for your valuable review. Before answering your questions, we want to clarify that RMSD is not the only metric for our Algorithm Selection model. We can use other metrics such as some scoring functions to replace RMSD when doing blind docking, and the result is still good. For example, we use part of the DUDE [1] dataset for blind docking, and evaluate the results based on AA-Score [2], a scoring function. Then, we build the AS model to select the optimal docking methods based on different protein and ligands settings. Here is the result:
>
> | Algorithm | Mean         | Std         | Median         | 0AA          | 5AA          |
> |-----------|--------------|-------------|-------------|--------------|--------------|
> | DiffDock  | 1.58317e+06  | 6.843969e+07| -6.14      | 75.384615 | 79.538462    |
> | GNINA     | 2.431891e+06 | 8.094182e+07| -7.82      | 96.102564 | 96.102564    |
> | SMINA     | 7.221649e+05 | 1.313745e+07| -9.07      | 97.179487 | 97.179487    |
> | TankBind  | 2.187797e+07 | 8.707069e+08| -1.19     | 51.384615 | 56.769231    |
> | DSDP      | 1.396005e+06 | 3.248848e+07| -9.07      | 83.897436 | 83.948718    |
> | GNNAS   | -3.988457e+00| 1.230933e+02| -9.22      | 98.256410 | 98.769231    |
> | best      | -9.578611e+00| 3.106873e+00| -9.66      | 99.179487 | 99.435897    |
>
> **Table 1:** AS model with DUDE dataset.
>
> For the AA-Score, the lower the value, the stronger the connection between docked results and receptors. From the Table 1, we can see that the results of Algorithm Selection is better than all the other single docking algorithms in terms of mean, median, a threshold of 0 and a threshold of 5, which means that the Algorithm Selection is practical in real blind docking situations. For the codes, we will release them soon.
>
> For the weakness and questions:
>
> 1.Separate GNNs for Proteins and Ligands: We designed separate GNN architectures for proteins and ligands due to the structural and functional differences between these two types of molecules. Ligands are not initially docked with proteins in our dataset, which requires a distinct analysis to accurately capture their individual properties before predicting interactions. This approach allows the models to learn specialized features of proteins and ligands independently, which are later combined to evaluate interactions effectively.
>
> 2.Selection of Atom Types: The choice of atom types was driven by their prevalence in our dataset and their commonality in drug discovery contexts. Heavier atoms, while important in certain specialized molecules, appear infrequently in our dataset. This decision was aimed at maintaining model efficiency and relevance to the most frequently encountered scenarios. If the dataset or application domain changes, we could expand the atom types considered in the construction of ligand graphs.
>
> 3.Incorporation of Edge Features: As mentioned in our paper, edge features are selected and used in our model. We incorporate bond types for ligands to reflect how atoms are connected and use the length of peptide bonds in proteins as edge features.
>
> 4.Comparison with Coarse-Grained Approaches: The approach of using amino acids to represent each node in a protein graph is a common method [3] that simplifies the complex structure of proteins while retaining essential biological information. This method differs from a coarse-grained approach. Essentially, the amino acid-based method provides a more detailed view compared to coarse-graining, which combines broader groups of atoms or even entire amino acid residues into single units, thus sacrificing some specificities for greater computational efficiency.
>
> 5.Algorithm Selection for GNN Models: The Algorithm Selection approach was not explicitly used for selecting GNN models in our current study, and it is not necessary. As demonstrated in table 1 of the paper, we have already shown that different GNN architectures could reach good performance for selecting docking methods for different protein-ligand pairs. For adjusting the GNN models, we could modify the layers without introducing additional computational by using a new AS framework.
>
> References:
>
> [1] Mysinger MM, Carchia M, Irwin JJ, Shoichet BK J. Med. Chem., 2012, Jul 5. doi 10.1021/jm300687e .
>
> [2] Pan, Xiaolin, et al. “AA-Score: A New Scoring Function Based on Amino Acid-Specific Interaction for Molecular Docking.” Journal of Chemical Information and Modeling, vol. 62, no. 10, 22 Apr. 2022, pp. 2499–2509, https://doi.org/10.1021/acs.jcim.1c01537. Accessed 22 Nov. 2024.
>
> [3] Jha, K., Saha, S. & Singh, H. Prediction of protein–protein interaction using graph neural networks. Sci Rep 12, 8360 (2022). https://doi.org/10.1038/s41598-022-12201-9

---

> > ### Comment · Reviewer_Jbng · 2024-11-25
> >
> > First of all, I would like to thank the authors for their responses. Benchmarking the AA-Score to measure protein-ligand favorability is a valuable addition. However, relying solely on the AA-Score is not sufficient. It is also very important to evaluate ligand quality, as it is just as critical as protein-ligand favorability in assessing overall performance.
> >
> >
> > **For Future Venues**
> >
> > As a suggestion for future work, I recommend focusing on making the model more efficient, accurate, and supported by more comprehensive benchmarks. This would significantly strengthen the paper for future submissions.

---

> ### Author Response · Authors · 2024-11-28
>
> We are grateful for the reviewer's thoughtful feedback and suggestions, which have provided us with valuable insights to further enhance our work. We particularly recognize the importance of evaluating ligand quality and appreciate your emphasis on this aspect. That being said, our study builds upon existing research and extends docking test scenarios to accommodate diverse datasets, as elaborated in our responses. Notably, our results demonstrate that GNNAS-Dock achieves superior performance across different metrics, while exhibiting improved robustness in various cases.
>
> We appreciate the reviewer's acknowledgement of the potential benefits of expanding benchmark sets and exploring alternative evaluation criteria. However, given GNNAS-Dock's capabilities, it is unlikely to be surpassed by standalone methods, unless a single approach consistently outperforms others across a wide range of scenarios. Theoretical and practical considerations, including the No Free Lunch theorem [1], suggest that such a scenario is improbable under fair experimental conditions.
>
> [1] D.H. Wolpert and W.G. Macready. No free lunch theorems for optimization. IEEE Transactions
> on Evolutionary Computation, 1(1):67–82, 1997.
>
> Furthermore, it is worth noting that GNNAS-Dock is a framework rather than a traditional docking method. This means it can be applied to any dataset, scoring function, or evaluation criteria, and can incorporate various docking algorithms as candidates for selection. Consequently, GNNAS-Dock has the potential to consistently outperform any standalone docking method, both existing and future ones, when they are included in the candidate algorithm set under any scoring function or evaluation criteria. The validity of GNNAS-Dock can be verified by comparing the best standalone docking algorithm's performance to the Oracle, where the best docking algorithm is selected for each docking scenario. If there is visible performance gap between the Oracle and a standalone, single best docking algorithm, it is likely that GNNAS-Dock will outperform all the docking algorithms in terms of overall, average performance.
>
> Going back to the reviewer comments, we are especially thankful for you to share that PoseBuster study which seems to be an important addition for molecular docking, indicating the weaknesses of some existing docking methods that released as the SOTA approaches against older and simpler docking methods. Thus, we already started following the idea proposed there on a larger set of docking scenarios coming from multiple datasets yet it will not be feasible to complete all the new experiments within this rebuttal period.

---

### Official Review · Reviewer_hyAZ · 2024-11-04

**Soundness:** 3
**Presentation:** 3
**Contribution:** 2
**Rating:** 3
**Confidence:** 4

**Summary:**

The paper introduces GNNAS-Dock, a novel GNN-based algorithm selection system designed to enhance molecular docking in blind docking scenarios. Recognizing that no single docking algorithm consistently outperforms others across all scenarios, a statement generally supported by the no free lunch theorem. The authors propose leveraging GNNs to predict the performance of various docking algorithms based on the structural data of ligands and proteins. They aim to predict the performance (measured in RMSD) of each candidate docking algorithm to identify the most accurate method for specific scenarios. This effectively allows to choose the most computationally efficient docking algorithm for each docking case, reducing computational time while maintaining high accuracy. Approach is validated on the PDBBind 2020 refined set, comprising approximately 5300 protein-ligand complexes. The candidate algorithms include a mix of traditional and deep learning-based methods: DiffDock, DSDP, TankBind, GNINA, SMINA, Qvina-W, and their combinations with p2rank, totaling nine algorithms. Proposed algorithm selection model achieves a mean RMSD of approximately 1.74A, significantly improving upon the best performing standalone docking algorithm, DiffDock, which has a mean RMSD of 2.95A. When considering computational efficiency, the model demonstrates a success rate of 79.7% in achieving an RMSD below the 2A threshold, with an average processing time of about 29 seconds per instance.

**Strengths:**

This work takes the first step in addressing a prominent challenge in computational drug discovery. The implementation of a GNN using standard ligand and protein features is effective and keeps the approach straightforward.
Moreover, the combination of an accuracy model with an efficiency model is beautifully done and in line with solving the performance-accuracy trade-off often faced in DD.

**Weaknesses:**

1. There's no need to emphasize the use of GNN or these standard features, as they are already well-established in this field. The development details of GNN-Lig and GNN-Prot might be more suitable for an appendix, allowing the main manuscript to focus on more critical analyses.
2. An ablation study could significantly enhance the work. For instance, exploring the impact of various features on docking algorithm selection would provide valuable insights into the interpretability and relevance of specific features across different approaches. Additionally, a deeper examination of the training datasets for machine learning models, as well as the testing systems used for physics-based methods, would strengthen the study's foundation.
3. Currently, the approach appears more like a baseline than a fully developed solution. This is understandable given the limited prior work in this area, as mentioned in the introduction and related work sections. However, enhancing GNN-Lig and GNN-Prot by incorporating richer features, such as learned sequence or structural representations (e.g., ESM or GearNet embeddings as node features), could throw further insights in how current approaches to represent ligands and proteins work for this problem.
4. Perhaps the most significant concern is the reliance on PDBBind for testing. While PDBBind is a valuable addition, it does not fully capture the complexities of docking scenarios. Many docking methods are trained on PDBBind, which includes systems traditionally well-suited for docking, thus limiting the ability to assess the model’s prospective potential. A carefully curated dataset with temporal splits would provide a more realistic evaluation of the model's performance across diverse scenarios.https://arxiv.org/abs/2308.09639 (as an example)

**Questions:**

1. Perform ablation studies on node features and incorporate richer representations.
2. Include temporal splits for the dataset used. Focusing on diverging from the dataset attained in training procedures and tunning of the ML models and docking scoring functions.

---

> ### Author Response · Authors · 2024-11-23
>
> We sincerely appreciate your recognition of our work as an initial step towards addressing the challenges in computational drug discovery. Below are our responses to address the weakness you pointed and answer your questions.
>
> 1.Mention of GNN and Standard Features: While GNN and standard features are well-established in the field, it is the first time GNN is applied for Algorithm Selection in molecular docking. We included a detailed discussion to ensure clarity for readers who may not be familiar with how these technologies can be used for Algorithm Selection in docking problems. However, based on your suggestion, we will consider moving detailed development descriptions of GNN-Lig and GNN-Prot to an appendix.
>
> 2.Ablation Study: We agree that an ablation study could significantly enhance the understanding of feature impact and model robustness. In the previous studies of Algorithm Selection in docking problems, some 2D feature importance analysis has been made. In Chen et al.’s study [1], they use algorithm selection for choosing an appropriate search box size for AutoDock 4. They mainly used regression-based algorithm selection model with manually extracted 2D features by Rdkit, and they found some features such as number of bond rotations, QED, and Kappa 1/2/3 to be important. We plan to include such studies to evaluate the impact of different node features on the performance of docking algorithm selection, which will also add some interpretability to our model. We have already conducted an ablation study by removing the coordinates of the residuals in protein graphs, and get a result in Table 1 below.
>
> | Algorithm | Mean    | Std     | Median     | 2A      | 5A      |
> |-----------|---------|---------|---------|---------|---------|
> | AS_no_coord  | 2.058863 | 4.445841| 0.700760| 80.840799| 90.075810|
> | AS_original  | 2.096518| 5.189157| 0.702726| 82.288077| 90.696072|
>
> **Table 1:** Ablation results of removing coordinates of the residuals.
>
>
> In the Table 1, AS_no_coord refers to the results of removing coordinates of the residuals in the protein graph representations, and AS_original refers to the result of using the graph representation introduced in our paper. We can see that the coordinates of proteins may not be an important node feature in the algorithm selection model, since it will not significantly affect the final result of AS model. We will also test for other node features as well.
>
> 3.Dataset and Evaluation Concerns:Your point about the reliance on PDBBind is well-taken. To address your concerns and prove the adaptability of AS model in more complex situations, we have tested the AS model in DUDE dataset [2]. We run blind docking in part of the DUDE dataset [3] with five different docking methods (DiffDock, TankBind, SMINA, GNINA, and DSDP) and uses AA-Score function for evaluating the docking results. The results of our algorithm selection model for choosing the appropriate docking methods in this situation is shown in Table 2.
>
> | Algorithm | Mean         | Std         | Median         | 0AA          | 5AA          |
> |-----------|--------------|-------------|-------------|--------------|--------------|
> | DiffDock  | 1.58317e+06  | 6.843969e+07| -6.14      | 75.384615 | 79.538462    |
> | GNINA     | 2.431891e+06 | 8.094182e+07| -7.82      | 96.102564 | 96.102564    |
> | SMINA     | 7.221649e+05 | 1.313745e+07| -9.07      | 97.179487 | 97.179487    |
> | TankBind  | 2.187797e+07 | 8.707069e+08| -1.19     | 51.384615 | 56.769231    |
> | DSDP      | 1.396005e+06 | 3.248848e+07| -9.07      | 83.897436 | 83.948718    |
> | GNNAS   | -3.988457e+00| 1.230933e+02| -9.22      | 98.256410 | 98.769231    |
> | best      | -9.578611e+00| 3.106873e+00| -9.66      | 99.179487 | 99.435897    |
>
> **Table 2:** AS model with DUDE dataset.
>
> For AA-Score, it is also the lower the better. In the result table, we can see that the Algorithm Selection model still outperforms all the single docking methods in terms of mean, median, a threshold of 0 and a threshold of 5. Also, it has lower variance, which means the algorithm selection reaches a stable result compared with other docking methods. This could demonstrate the applicability and adaptability of algorithm selection model in more complex docking situations.
>
> References:
>
> [1] Chen, T., Shu, X., Zhou, H. et al. Algorithm selection for protein–ligand docking: strategies and analysis on ACE. Sci Rep 13, 8219 (2023). https://doi.org/10.1038/s41598-023-35132-5.
>
> [2] Mysinger MM, Carchia M, Irwin JJ, Shoichet BK J. Med. Chem., 2012, Jul 5. doi 10.1021/jm300687e .
>
> [3] Pan, Xiaolin, et al. “AA-Score: A New Scoring Function Based on Amino Acid-Specific Interaction for Molecular Docking.” Journal of Chemical Information and Modeling, vol. 62, no. 10, 22 Apr. 2022, pp. 2499–2509, https://doi.org/10.1021/acs.jcim.1c01537. Accessed 22 Nov. 2024.

---

### Meta-Review · Area_Chair_oG6e · 2024-12-21

**Metareview:**

(a) The paper proposes GNNAS-Dock, a GNN-based algorithm selection system for molecular docking. It aims to predict the performance of different docking algorithms and select the most suitable one in terms of accuracy (using RMSD) and computational efficiency. The findings show that the proposed model achieves a better mean RMSD compared to the best-performing standalone docking algorithm (DiffDock) and has a certain success rate in achieving RMSD below a threshold while maintaining efficiency.

(b) The strengths of the paper include addressing a practical problem in computational drug discovery and applying GNNs for algorithm selection. The combination of accuracy and efficiency models is also considered reasonable.

(c) The weaknesses: The use of GNN and standard features is not novel enough, and detailed development descriptions might be better placed in the appendix. There is a lack of ablation study and the approach seems more like a baseline. The reliance on PDBBind for testing is a concern as it may not fully represent docking complexities. The paper also has issues such as using an outdated metric (RMSD), potential information leakage, insufficient explanation of data preparation and training, and lack of physicochemical validation.

(d) The main reasons for rejection are that the paper has many methodological flaws. The results may not be generalizable due to the evaluation on a single dataset with potential information leakage. The RMSD metric is not sufficient to evaluate the docking quality comprehensively. The paper also lacks comparison with other algorithm selection methods and has insufficient analysis of the model's predictions. Additionally, the construction of the neural network and the choice of dataset raise concerns about the model's robustness and relevance.

**Additional Comments On Reviewer Discussion:**

Reviewers raised several points including the novelty of using GNN, the adequacy of evaluation metrics (RMSD), the dataset used (PDBBind), the construction of neural networks, and the lack of comparison with other algorithm selection methods. The authors addressed these points by considering moving GNN development details to the appendix, testing on an additional dataset (DUDE), clarifying the graph construction and dataset usage, and discussing the possibility of future comparisons with other algorithm selection methods. However, the authors' responses did not fully address the concerns about the fundamental flaws in the methodology and the lack of comprehensive evaluation. The additional experiments on the DUDE dataset did not convince the reviewers that the model's performance was reliable and generalizable. Overall, the weaknesses still outweighed the strengths, leading to the decision to reject the paper.

---

### Decision · Program_Chairs · 2025-01-22

Reject